# 3D-Printed Tubular Scaffolds Decorated with Air-Jet-Spun Fibers for Bone Tissue Applications

**DOI:** 10.3390/bioengineering9050189

**Published:** 2022-04-27

**Authors:** Febe Carolina Vazquez-Vazquez, Daniel Chavarria-Bolaños, Marine Ortiz-Magdaleno, Vincenzo Guarino, Marco Antonio Alvarez-Perez

**Affiliations:** 1Laboratorio de Materiales Dentales, DEPeI, School of Dentistry, Circuito Exterior, s/n, Ciudad Universitaria, Mexico City 04510, Mexico; fcarolina.vazquez@gmail.com; 2Facultad de Odontología, Universidad de Costa Rica, San Jose 11501-2060, Costa Rica; danielchava2@gmail.com; 3Faculty of Stomatology, Autonomous University of San Luis Potosi, San Luis Potosi 78000, Mexico; marine.ortiz@uaslp.mx; 4IPCB/CNR, Institute of Polymers, Composites and Biomaterials, Consiglio Nazionale delle Ricerche, Mostra D’Oltremare, Pad. 20, V. le J.F. Kennedy 54, 80125 Naples, Italy; 5Tissue Bioengineering Laboratory, DEPeI, School of Dentistry, Universidad Nacional Autonoma de Mexico (UNAM), Circuito Exterior s/n C.P., Mexico City 04510, Mexico; marcoalv@unam.mx

**Keywords:** additive manufacturing, tubular scaffolds, nanofibers, bone, biocompatibility, tissue engineering

## Abstract

The fabrication of instructive materials to engineer bone substitute scaffolds is still a relevant challenge. Current advances in additive manufacturing techniques make possible the fabrication of 3D scaffolds with even more controlled architecture at micro- and submicrometric levels, satisfying the relevant biological and mechanical requirements for tissue engineering. In this view, integrated use of additive manufacturing techniques is proposed, by combining 3D printing and air-jet spinning techniques, to optimize the fabrication of PLA tubes with nanostructured fibrous coatings for long bone defects. The physicochemical characterization of the 3D tubular scaffolds was performed by scanning electron microscopy, thermogravimetric analysis, differential scanning calorimetry, profilometry, and mechanical properties. In vitro biocompatibility was evaluated in terms of cell adhesion, proliferation, and cell–material interactions, by using human fetal osteoblasts to validate their use as a bone growth guide. The results showed that 3D-printed scaffolds provide a 3D architecture with highly reproducible properties in terms of mechanical and thermal properties. Moreover, nanofibers are collected onto the surface, which allows forming an intricate and interconnected network that provides microretentive cues able to improve adhesion and cell growth response. Therefore, the proposed approach could be suggested to design innovative scaffolds with improved interface properties to support regeneration mechanisms in long bone treatment.

## 1. Introduction

The design of instructive scaffolds able to satisfy biological, mechanical, and morphological requirements is currently an important issue for bone tissue engineering [1]. To date, bone is the second most transplanted tissue worldwide; more than 4 million surgical procedures are performed annually on bone tissue. Many bone defects are regenerated with bone grafts, but there are still substantial clinical problems associated with autologous bone grafts and surgical techniques [2].

Therefore, numerous studies have suggested the design of three-dimensional structures, also known as scaffolds [3], with different microstructural characteristics but with the same purpose: to create biomechanical support that allows the reproduction of the extracellular microenvironment of bone tissue. This may be reached through a sapient manipulation of biomaterials via processing techniques tailored to confer them peculiar properties at different size scales that reply to that of natural tissue as a function of the anatomical location and position [4]. For instance, fine control of structural porosity in terms of characteristic pore size and interconnections is crucial to assure the proper exchange of oxygen nutrients and metabolites to induce and regulate cellular behavior (i.e., migration, cell integration) that are also involved in the osteointegration mechanisms [5,6].

Moreover, a scaffold’s micromorphological design must be coupled to an accurate definition of the matrix chemical composition to reach the biological targets [7]. In vivo and in vitro studies have recently reported enhanced healing and extensive bone formation defects through the incorporation of immobilized peptides, proteins, and biomolecules [8] as growth factors [9], vascular endothelial growth factor (VEGF) [10], morphogenetic proteins, and platelet-derived growth factor, to improve on osteoconductivity, osteoinductivity, and osteogenesis [11]. In this view, scaffold fabrication technologies must address the facile incorporation of biologically active components to be therapeutically delivered during the regeneration process [12].

In the last two decades, a wide range of processing methodologies have been variously investigated to provide the fabrication of 3D scaffolds in different forms (i.e., particles, sponges, blocks, spheres, nanofibers, tubes) that, to some extent, might mimic the micro- and/or nanostructure of the extracellular matrix, also reproducing similar functionalities of the native tissue [13,14,15]. They vary from casting or layer-by-layer deposition methods conventionally used for the fabrication of 2D scaffolds [16,17] to gas foaming, supercritical CO_2_, and freeze-drying suitable for the fabrication of 3D foams [18,19,20], until more recent additive manufacturing approaches based on rapid prototyping and electro fluid dynamics techniques [21,22] mainly addressed to the design of highly customized architectures for personalized medicine.

In this context, 3D printing is currently one of the most innovative technological procedures to design highly complex and hierarchical architectures. A simple, fast, and easy process can manufacture various substitutes for tissues and organs that have entirely or partially lost their function [23,24,25]. Synthetic polymers are elective materials for using the 3D printing technique; among them, polylactic acid (PLA), a linear aliphatic polyester that can be obtained from renewable resources at relatively low costs, possesses peculiar properties in terms of adsorbability and nontoxicity after degradation, making it particularly demanding for the manufacturing of 3D-printed scaffolds [26]. Moreover, it has been approved by the Food and Drug Administration, making it suitable for all the applications that involve direct contact with biological fluids [27].

In recent years, scaffolds with tubular geometry have been variously investigated, exhibiting interesting results in vascular, nerve [27,28,29], tendon–bone junction [30], and, more in general, organ regeneration [31]. Indeed, tubular microstructure properly offers optimal external surfaces for cell attachment and internal surfaces for cell infiltration. Indeed, tubular morphology permits the cells to adhere around the circumference of the construct, occupying a larger surface area that is in contact with the surrounding tissue of the bone defect. Moreover, this peculiar geometry also can be loaded with microparticles, nanoparticles, hydrogel, porous, and fiber–matrix for therapeutic approaches via the controlled release of bioactive molecules [32]. For this purpose, tubular scaffolds may efficiently function as a growth guide for the regeneration of peripheral nerve [33] or long bone defects [34,35].

However, the main constraint of 3D printing still concerns the limited capability to impart a controlled nanoscale roughness on the surfaces at the interface with cells. In our recently published paper, we suggested the fabrication of a tubular scaffold by an additive manufacturing approach based on the combination of two processing techniques: 3D printing and the air-jet spinning technique (AJS). Preliminary results confirmed the scaffold’s biocompatibility, suggesting them as a candidate for use in tissue engineering. However, the small sizes of the tubes make them not applicable for the regeneration of long bone defects [36]. Herein, we further implement the proposed additive manufacturing approach by optimizing the fabrication of 3D-printed PLA tubes with geometry tailored to the long bone defect and decorating the external surface with air-jet-spun PLA fibers to improve the cell interface. This work characterized mechanical, thermal, and surface topographical properties, and biocompatibility tests were performed using human fetal osteoblasts (hFOB) to validate their use as a bone growth guide.

## 2. Materials and Methods

### 2.1. Fabrication

The synthesis of the 3D tubular scaffold started with the design of cylindrical geometry using the Cura 3.0 software. Then, the scaffold was fabricated by additive manufacturing using a 3D printer (Maker-Mex, Mexico) by using polylactic acid filament (PLA, 0.75 mm). The printed tubular scaffold dimension was 9 mm in diameter and 35 mm in length. The layer coating functionalization by fiber-spun mats was carried out with air-jet spinning (AJS). Briefly, PLA pellets (C_3_H_6_O_3_; MW 192,000 from NatureWorks D2002) were dissolved in chloroform/acetone (volume ratio of 3:1) to obtain a 7% (*w*/*v*) polymer solution. Then, the polymeric solution was placed in commercially available ADIR model 699 airbrushes with a 0.3 mm nozzle diameter and a gravitational feed of the solution to synthesize the fiber membrane scaffold. The airbrush was connected to a pressurized argon tank (CAS Number: 7740-37, concentration > 99%, PRAXAIR, Mexico), and a pressure of 35 psi and 10 cm from the nozzle to the 3D tubular scaffold were constantly maintained for fiber deposition.

### 2.2. Physicochemical Characterization

The structure and morphology of the printed scaffold were examined using a scanning electron microscope (JEOL JSM-6700F microscope) with a 5 kV acceleration voltage for the electron beam. The samples were sputter-coated with a 5 nm thin gold layer (EMS 150R, Quorum, East Sussex, UK) to analyze the outer and inner surfaces. In addition, specific observations were performed to examine the interphase between nanofibers and the core printed material.

The printed scaffold’s mechanical characterization was analyzed by testing the compressive strength and Young’s modulus using a universal testing machine INSTRON (model 1125, Norwood, MA, USA), where the crosshead was moved at a speed of 1.3 mm/min until the specimen failed, and the load cell was 5000 N. Tests were conducted on 8 specimens for each sample type.

The topographic images were taken with a contact profilometer (Bruker Dektak TX, Billerica, MA, USA). A 2 μm radius needle tip was used. The sweep axis was taken with an applied force of 4 mg, and the resolution was adjusted to 0.033 μm/point (corresponding to a needle tip speed of 10 μm/s covering a total area of 500 × 500 μm). The image was constructed by joining 500 scans with a separation of 1 μm. The roughness values were evaluated using the entire measured area to determine the average roughness (Ra) and the average maximum depressions (Rq).

Thermogravimetric analysis (TGA) was performed using TGA-Q500 equipment (TA Instruments, New Castle, DE, USA). Platinum baskets were tared before automatically weighing 4–6 mg of the sample to be analyzed. After loading, the furnace was closed and running on a ramp from 25 °C to 1000 °C, with a heating rate of 10 °C/min, using a nitrogen flow rate of 90 mL/min. Data were analyzed using TGA software Universal Analysis Version V4.5A (TA Instruments, New Castel, DE, USA) to identify onset points (To), inflection points (Tp), and maximum mass loss point (Tmax).

Differential scanning calorimetry (DSC) (Q200, TA Instruments, New Castle, DE, USA) was performed with 2.5–3 mg of each sample, running a ramp from 25 °C to 250 °C, with a heating rate of 10 °C/min. Glass transition temperature (Tg) and melting points (Tm) were calculated. The results were obtained from the first warm-up run.

### 2.3. In Vitro Studies

Human fetal osteoblast cells (hFOB, 1.19 ATCC CRL-11372) were used to evaluate the biological response of the tubular printed and coating 3D scaffold. hFOB cells were cultured in 75 cm^2^ cell culture flasks containing a 1:1 mixture of Ham’s F12 medium Dulbecco’s Modified Eagle Medium (DMEM, Sigma-Aldrich, St. Louis, MO, USA), supplemented with 10% fetal bovine serum (FBS, Biosciences, Princeton, NJ, USA), 2.5 mM L-glutamine and antibiotic solution (streptomycin 100 μg/mL and penicillin 100 U/mL, Sigma-Aldrich). The cell cultures were incubated in a 100% humidified environment at 37 °C in 95% air and 5% CO_2_. hFOB on Passages 2–6 were used for all the experimental procedures. Before the biological assays, a tubular printed and coating 3D scaffold was placed in a 24-cell culture plate and sterilized by immersion in 70% of ethanol (*v*/*v*) with an antibiotic solution (streptomycin 100 μg/mL) penicillin 100 U/mL) for 30 min. After sterilization, scaffolds were rinsed with PBS, distilled water three times, and air-dried.

To evaluate the cell adhesion of the hFOB onto the tubular printed and coating 3D scaffolds, cells were seeded at 1 × 10^4^ cells/mL and allowed to adhere to standard cell culture for 4 and 24 h. After the prescribed time, the tubular scaffolds were rinsed three times with PBS to remove nonadherent cells. Then, adherent cells onto the scaffolds were fixed with 4% paraformaldehyde and incubated with 0.1% Crystal Violet solution for 15 min. Then, the dye was extracted with 0.1% sodium dodecyl sulfate (SDS), and optical absorption was quantified by spectrophotometry at 545 nm with a plate reader ChroMate (Awareness Technology, Palm City, FL, USA).

Cell viability of hFOB onto the tubular printed and coating 3D scaffolds was analyzed after 3, 5, 7, and 9 days of culture. First, the viability was checked by the WST-1 assay based on the ability of the mitochondrial succinate-tetrazolium reductase of living cells to reduce a WST-1 salt (4-(3-(4-iodophenyl)-2-(4-nitrophenyl)-2H-5-tetrazolium)-1,3-benzene disulfonate) to produce a water-soluble formazan dye product. The concentration of the formazan product is directly proportional to the number of metabolically active cells. hFOB at 1 × 10^4^ cells/mL seeded onto the tubular scaffold at the prescribed times were washed with PBS and incubated with 400 μL of fresh culture medium containing 40 μL of the cell proliferation reagent WST-1 for 4 h at 37 °C. Then, 200 μL of the supernatant was removed, and absorbance was quantified by spectrophotometry at 450 nm with a ChroMate plate reader. During the experimental time, the culture medium was exchanged every third day.

The cell–material interaction of hFOB cells seeded at 1 × 10^4^ cells/mL onto a tubular scaffold was examined using SEM. For SEM analysis, the tubular scaffolds were washed three times with PBS, fixed with 2% glutaraldehyde for 1 h, and then dehydrated with a graded series of ethanol (25–100%); finally, the samples were subjected to critical point drying. Finally, the samples were sputter-coated with a thin layer of gold–palladium and examined by SEM.

### 2.4. Statistical Analysis

All quantitative data were expressed as the average ± standard error of the mean. Numerical data were analyzed via the Student’s *t*-test to determine differences among the groups. Statistical significance was considered at *p* < 0.05.

## 3. Results and Discussion

### 3.1. Morphological Characterization

It is well known that a predominance of a longitudinal axis characterizes long bones; therefore, a tubular scaffold made of a polymer can create a rigid structure that fulfills the effect of acting as a guide axis for the formation of the tissue on its external and internal surface [34]. It has been reported that tubular scaffolds are similar to long natural bones [35], by mimicking the extracellular matrix (ECM) in terms of hierarchically organized mineralized collagen fibers. Thus, in this study, we proposed fabricating a 3D bilayered scaffold by an additive manufacturing strategy involving the 3D printing technique to impart the tubular architecture and AJS technology to improve surface properties by controlled deposition of nanofibers (Figure 1).

In Figure 1A, the macroscopic aspect of the 3D-printed tube is reported. It is presented as a rigid structure with a homogeneous thickness and a milky color. An accurate design of the printing pattern enabled it to accurately control wall thickness but preserve the peculiar tube geometry without macroscopic defects (Figure 1A). Secondly, the addition of the fibrous coating makes the tube surface highly homogeneous and form a circumferential layer of uniformly spatially distributed fibers suitable to improve the interface properties with cells (Figure 1B). In agreement with previous studies, the decoration of 3D scaffolds via nanofibers with a high specific surface area is recognized as a valid strategy to reproduce the fibrillar structure of the bone tissue ECM, basically providing nanoretentive spaces among nanofibers that are recognized by cells as preferential anchor points to initiate the biological process of cell adhesion [37,38,39].

Figure 2 shows SEM micrographs taken from tubular scaffolds without fibrous coating: the morphology of the layer-by-layer structure is characterized by the presence of thin PLA layers, formed during the melt polymer solidification, that concur to locally form some reliefs, crests, and depressions at the proximity of the strut surface (Figure 2A,D). High-magnification images confirmed this evidence, remarking the prevalence of smooth surfaces of the outer surface (Figure 2B,C) and the formation of depression and pores mainly along the inner surface of the scaffold (Figure 2E,F). Accordingly, these peculiar topographical signals are suitable to improve the interactions with cells in vitro. In the presence of fiber decoration, smooth surfaces of the outer scaffold side support the formation and the attachment of a homogeneous pattern of random fibers, which tend to form local fiber bundles with different densities and average diameters of 0.430 ± 0.205 μm (Figure 3A–C). This is a consequence of accurate control of the pressure during the fiber deposition that allows guiding the evaporation of solvents to obtain stably adherent fiber meshes onto the surrounding 3D-printed tubes. Of note, the fiber coating adhered to the 3D-printed structure, seen from the inner side of the tube, tends to form interwoven meshes that resemble the fibrillar structure of the ECM, assuring better interaction of the cells at the interface with the tubular scaffold, which could be essential for cell growth and tissue regeneration (Figure 3D–F).

### 3.2. Thermal Response

To analyze the thermal stability of the tubular scaffolds, TGA was employed to monitor the weight change of the scaffolds in a controlled atmosphere and determine the decomposition temperature of the tubular scaffolds. In addition, the onset point (To), the inflection point (Tp), and the mass loss temperature (Tmax) of the PLA filament and the tubular scaffolds with and without NF coating were determined. Tp was determined by calculating the peak of the first derivative of the weight loss curve. The PLA filament obtained the highest temperature of To. To achieve a 10% mass loss, the PLA filament required 350.52 °C, while the PLA 3D tubule at 311.42 °C and the 3D scaffold functionalized with PLA nanofiber coating at 297.74 °C; thus, the tubule with fibers obtained the lowest temperature value for the weight loss (Figure 4A). The temperature at which 90% of the mass was lost was achieved at 367.65 °C, 364.31 °C, and 347.69 °C for the PLA filament, the tubule without fibers, and the tubule with the fiber-spun coating mat, respectively (Figure 4A). The maximum mass loss rate or Tp was higher in the PLA filament, achieved at 361.19 °C. The uncovered PLA tubule showed a Tp of 357 °C, and the tubule covered with PLA nanofibers had a Tp of 338.16 °C (Figure 4B). The tubular structure disseminates heat differently from the solid structure of the PLA filament; on the other hand, the behavior of the tube with fibers is attributed to the fact that the fibers allow better distribution of the temperature, meaning that, To starts at a lower temperature, and Tp is achieved sooner. The faster melting of the PLA fibers may explain why the 90% mass loss was achieved sooner in the functionalized scaffold, while the pure PLA filament required 20 °C to achieve the same mass loss. This study revealed that coating with fiber-spun mat provides polymeric tubules with improved temperature distribution, diminishing the temperature to achieve mass loss.

The results of DSC evaluated the glass transition temperature (Tg) of the PLA filament, which was 62.01 °C, in the tubule without coating 61.12 °C, and for the tubule with the fiber-spun coating mat 61.93 °C. As a result, the melting temperature (Tm) for the PLA filament was 169.40 °C, for the tubule without fibers 167.87 °C, and for the tubule with the coating, the fibers were 167.98 °C (Figure 5). It is important to highlight that similar Tg may explain the homogeneity obtained when the polymer is either printed or added by air-jet spinning over the printed scaffold. At this thermal range, the polymer behaves like a whole structure. Another important observation is a slight shift of the exothermic crystallization signal after Tg was achieved. The fibrillar structures over the scaffold improved this crystallization when the temperature was increased; thus, the PLA filament requires a slightly higher temperature to complete this process. This enhanced behavior may explain why printed, and printed scaffolds covered with PLA nanofibers showed a slight diminishing in the Tm. This behavior was previously reported in a similar scaffold design [36].

### 3.3. Profilometry

Profilometry analyzed the surface parameters topography of the tubular scaffold to obtain the mean roughness (Ra) values because it allows larger surface scans and ease of use. The data obtained by profilometry will enable us to establish significant differences between the tubular scaffold without the coating with an Ra value of 0.028 µm and for the tubule with the presence of the fiber-spun coating was 0.2 µm. This indicates that the tubular scaffold with fiber-spun mat has a surface with higher roughness than the tubular scaffold without it, observing ridges and valleys, which coincides with the surface characteristics of the microphotographs (Figure 6).

### 3.4. Mechanical Characterization

The compressive strength and Young’s modulus analysis were performed to analyze whether the combination of a tubular scaffold with the presence of the fiber-spun mat provides better mechanical and physical characteristics than the tubular scaffold without it. As a result, the tubular scaffold had a value of 141.8 ± 19.39 MPa in the stress test, and the tubular scaffold with the fiber-spun coating mat had a value of 150 ± 20.91 MPa (Figure 7A); thus, the scaffold behaved as a semi-crystalline polymer, indicating that the polymer chains are intercrossed and amorphous, permitting a reversible deformation and higher resistance to fracture, the latter coinciding with the tests performed by X-ray diffraction reported [36,40].

The Young’s modulus in the tubular scaffold was 839.8 ± 41.63 MPa, and that of the tubular scaffold with the fiber-spun coating mat was 825.1 ± 32.52 MPa. However, there was no statistically significant difference between them (Figure 7B). Although no significant differences were obtained between tubules with and without fibers in the mechanical tests, PLA exhibits a thermoplastic behavior that improves its resistance to deformation and, consequently, its resistance to fracture. It has been reported that there is a relationship between scaffold morphology, components, and fracture strength in materials used to functionalize scaffolds [41]. Studies have reported increased fracture strength values and the improvement of mechanical characteristics in general by incorporating nanoparticles and fibers, providing more excellent thermal stability, a better coefficient of thermal expansion, and more excellent resistance to decomposition [42,43]. Regarding the mechanical properties of the cylindrical or tubular scaffolds, they are sufficiently robust for structural support in long bones; they improve their osseointegration due to that the internal part of the tubular structure is hollow, and the formation of tissue will be in both directions: from the outer part of the tubular scaffold to the internal part and vice versa.

### 3.5. Cellular Adhesion and Proliferation

The biocompatibility of the scaffold material depends on that it does not promote any cytotoxic response and promotes cellular adhesion. Moreover, if the scaffold offers the conditions for the cells to anchor, adhere, and proliferate, the surface induces a specific cue that enhances cellular processes and regulates extracellular matrix deposition. In this manner, cellular behavior, response, and function will depend on the biological and morphological microenvironment created by the scaffold’s surface properties, composition, and morphology [44]. In this study, we employed PLA polymer to produce both tubular and fiber coating of the surface scaffold. PLA is a widely studied material used in tissue engineering to manufacture scaffolds in human body tissues. Our results of the printed tubular surface of 9 mm in diameter allow the adhesion of the hFOB cells with significant differences (*p* < 0.05) between the PLA filament and 3D tubular scaffold with respect to the 3D tubular coating with fiber-spun mat at 4 and 24 h (Figure 8A). Cellular adhesion results indicate that tubular possessed a suitable substrate for cell response; we evaluated cell proliferation. Our results revealed a continuous increase in cell viability at 3, 5, 7, and 9 days in the PLA filament and the printed tubular scaffold. However, the growth of hFOB cells was higher in the printed tubular scaffold coating with the fiber-spun mat during all evaluation periods, with a statistically significant difference (*p* < 0.05) at 7 and 9 days (Figure 8B). The results indicate that coating on the tubular scaffold regulates the adhesion, proliferation, and viability of hFOB cells because the fibers mesh works as a platform cue that improves cell biocompatibility and supports colonization and cell growth. Our results are consistent with previous studies that have shown that cellular behavior is sensitive to the surface topography of the scaffold, demonstrating that changes in the surface microstructure could affect cell behavior and also that the microarchitecture offered by nano- and microfibers favors cellular anchorage because it creates multiple sites for protein uptake and promotes binding sites providing sufficient space for cell growth and improve the oxygen supply to the large graft [24,26,45,46].

### 3.6. Cellular Morphology

The additive manufacturing strategy to printed tubular rigid scaffold showed that hFOB cells colonize the surface of the convex wall. The cell could be observed as clusters in the topography of the tubular scaffold. Furthermore, the magnification could be observed as the junction between the cells and the layers formed by the scaffold that serve as retention spaces for cell attachment (Figure 9A,B). Compared to the tubular scaffold coating with fiber-spun mat, hFOB cells covered the surface attached as cell–cell conglomerates with elongated and slightly flattened morphologies (Figure 9C,D). Moreover, the cytoplasmic prolongations of cells and lamelopods interacting with the neighboring cells are observed in both cases. This response agrees with studies showing that interconnected pores and large formed by fiber mats facilitate cell adhesion and cell–cell communications to interact directly with the fibers [43]. In contrast, the diameter of the fiber has been linked to both proliferation and cell morphology and is probably involved in offering a favorable microenvironment that regulates and maintains the cell phenotype and growth based on the orientation of the fibers [47,48,49].

## 4. Conclusions

We proposed an integrated additive manufacturing strategy to fabricate PLA tubular scaffolds decorated with airbrush-spun nanofibers for bone tissue engineering applications. Our results showed that the 3D scaffold had stable mechanical and thermal properties, and the fiber coating provided microretentive surface cues for improving the cell’s biological response. Therefore, we suggested that the proposed approach produces bio-inspired scaffolds loaded with growth factors or other bioactive molecules for therapeutic administration during the regeneration process. In the near future, further studies will be conducted to load bioactive phases (i.e., calcium phosphates) into the scaffold struts to investigate the osteoinductive properties, biomineralization, and differentiation process of 3D composite scaffolds.

## Figures and Tables

**Figure 1 bioengineering-09-00189-f001:**
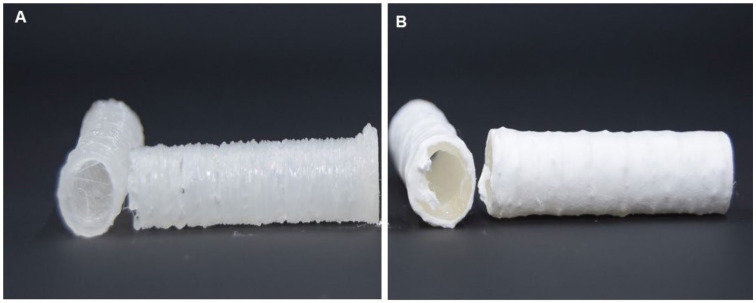
Macroscopic images of the 3D tubular scaffolds by additive manufacturing: 3D printed tubes (**A**) and fiber coated tubes (**B**).

**Figure 2 bioengineering-09-00189-f002:**
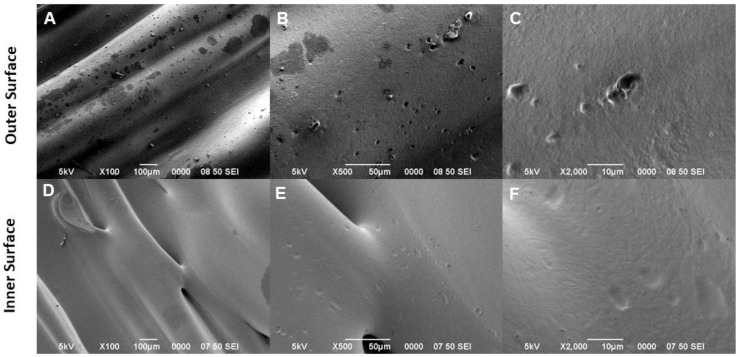
SEM micrographs showing the outer and inner morphology of the 3D tubular printed scaffold. The polymer layers are observed with reliefs and depressions in the outer (**A**) and inner structure (**D**) which become smooth and with some depressions in its topography. (**B**,**E**) At high magnification on both sides and the surface continues to be smooth (**C**,**F**).

**Figure 3 bioengineering-09-00189-f003:**
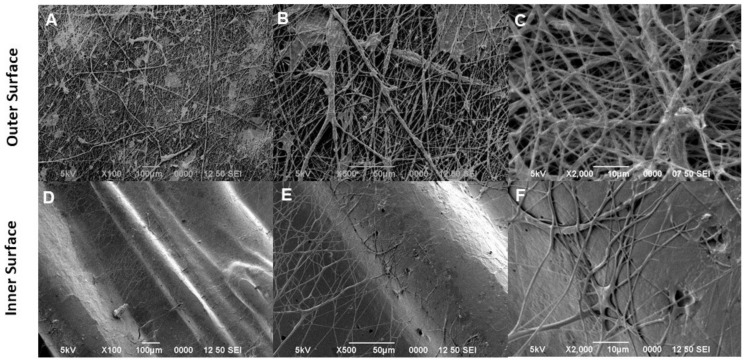
SEM images showing the outer and inner morphology of the 3D tubular printed scaffold coating with fiber-spun mat by air-jet spinning. The outer surface showed a fiber network covering the external surface of the tubule in random interconnection (**A**–**C**). The inner surface showed the few fibers covering the depressions (**D**) and at higher magnifications showed the random interconnections of the fibers forming an interfibrillar porosity (**E**,**F**).

**Figure 4 bioengineering-09-00189-f004:**
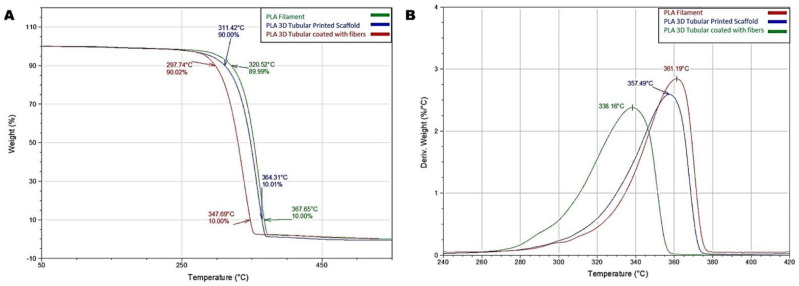
Thermogravimetric analysis (TGA) curves of the PLA filament and the tubular scaffolds with and without the fiber-spun coating layer. (**A**) Weight loss (%) in the function of temperature change. (**B**) First derivate weight loss (%/°C) in the function of temperature change.

**Figure 5 bioengineering-09-00189-f005:**
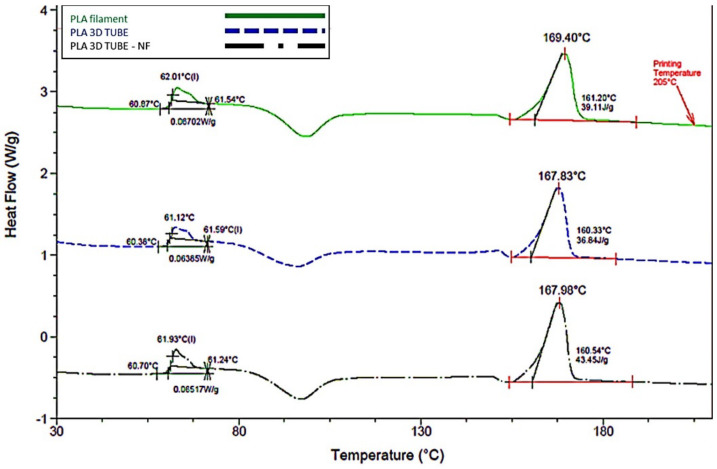
DSC thermograms of the PLA filament and the tubular scaffolds with and without the fiber-spun coating layer.

**Figure 6 bioengineering-09-00189-f006:**
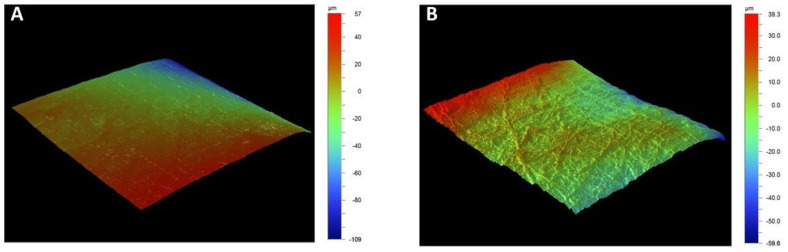
Optical profiler data showing the topography of the 3D tubular printed scaffold (**A**) and 3D tubular printed scaffold coating with fiber-spun mat by air-jet spinning (**B**).

**Figure 7 bioengineering-09-00189-f007:**
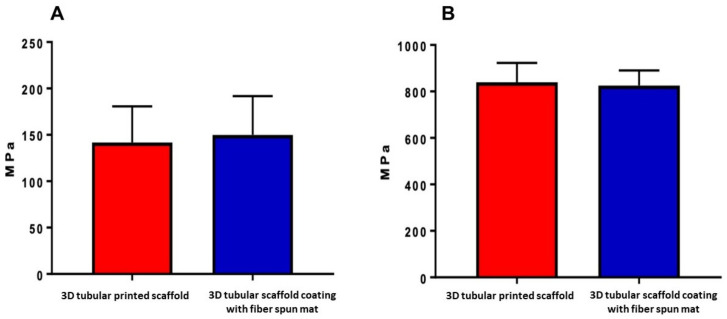
Mechanical properties of the 3D tubular printed scaffold with and without the fiber-spun coating mat. (**A**) Strength test of the 3D tubular scaffold. (**B**) Young’s moduli of the 3D tubular scaffold.

**Figure 8 bioengineering-09-00189-f008:**
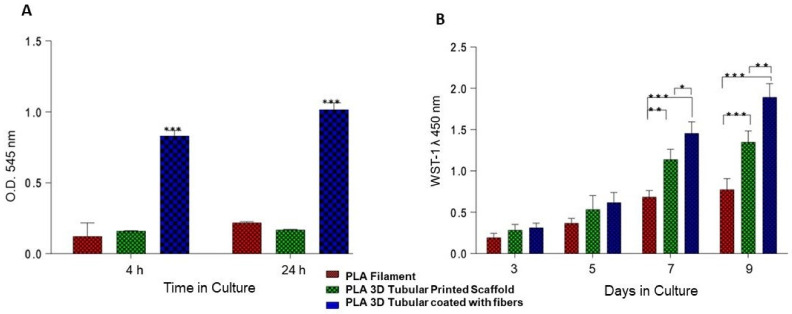
Cell adhesion of hFOB at 4 and 24 h of cell culture time in the 3D tubular scaffold (**A**). Cell viability of hFOB at 1, 3, 5, and 9 days of culture with the tubular scaffolds (**B**). Statistical significance is indicated by an asterisk (*, **, ***) *p* < 0.05.

**Figure 9 bioengineering-09-00189-f009:**
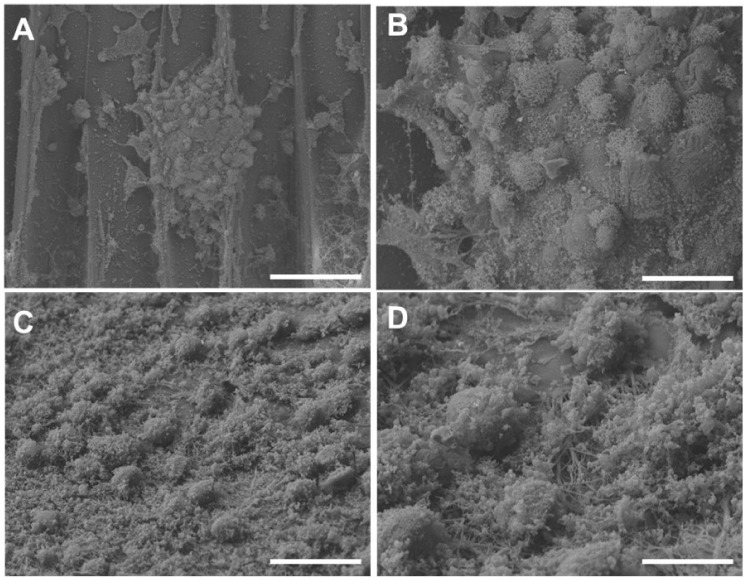
Images obtained with SEM of hFOB cells in the 3D tubular scaffold after 24 h of culture. Attached cells and conglomerated cells on the surface of the 3D tubular scaffold could be observed (**A**,**B**). Tubular scaffold coating with fiber-spun mat showed hFOB cell-attached and slightly flattened morphology on the surface (**C**,**D**). Bar = 10 μm.

## Data Availability

All datasets generated for this study are included in the article.

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
