# Peer review of "3D-Printed Tubular Scaffolds Decorated with Air-Jet-Spun Fibers for Bone Tissue Applications"

_bioengineering, 2022, doi:10.3390/bioengineering9050189_

Round 1

Reviewer 1 Report

Dear,

The authors developed scaffolds for application in tissue engineering. The manuscript is interesting and therefore has merit for publication. However, some corrections must be made:

1°) Introduction needs recent references. Please update, especially in the 2020-2022 range.

2°) In the introduction please expand the reason for using PLA. The comment was brief.

3°) Please add the commercial code of the PLA used. In addition, density, melt flow index (MFI), etc.

4°) Fabrication – Authors could add an illustrative schematic of the experiment. This improves the understanding of the experimental procedure.

5°) Page 4. Line 128-129. Correct the indent speed. The assay average was with how many samples? add to manuscript

6°) TG. Line 139-140: Correct the temperature (use point). What gas flow is used? What gas is used? Add to manuscript. Why did you use a rate of 20°C/min? The default is 10°C/min

7°) DSC. Are the results from the first or second warm-up run? Inform in the manuscript.

8°) Page 9. Please correct the units from “Mpa” to “MPa”.

Author Response

Please, see the attachment for the replies to the comments.

Reviewer 2 Report

I have reviewed the paper "Air Jet-spinned fibres decorated tubular 3D printed scaffolds for bone tissue applications" and found the paper can be accepted after major revision.

-More physical explanation of results is required.

-What is the reason for using this technique to make bone scaffolds?

-The Abstract should be improved.

-The quality of the figures should be improved.

-What is the study of mechanical and biological properties of this type of scaffold in comparison with other researches?

-Finally, the language of the paper needs to be polished.

-Based on the topic the title is so short and needs to be clarify the problem statement clearly.

-The Figures quality are too weak please improve the quality and put some arrays on the important part

-The language of the paper needs major polish.

The following reference are introduce to compare for the preparation of nanocomposite.

-Khandan, A., Abdellahi, M., Ozada, N., & Ghayour, H. (2016). Study of the bioactivity, wettability and hardness behaviour of the bovine hydroxyapatite-diopside bio-nanocomposite coating. Journal of the Taiwan Institute of Chemical Engineers, 60, 538-546.

-Sharafabadi, A. K., Abdellahi, M., Kazemi, A., Khandan, A., & Ozada, N. (2017). A novel and economical route for synthesizing akermanite (Ca2MgSi2O7) nano-bioceramic. Materials Science and Engineering: C, 71, 1072-1078.

-Salmani, M. M., Hashemian, M., & Khandan, A. (2020). Therapeutic effect of magnetic nanoparticles on calcium silicate bioceramic in alternating field for biomedical application. Ceramics International, 46(17), 27299-27307.

-Kazemi, A., Abdellahi, M., Khajeh-Sharafabadi, A., Khandan, A., & Ozada, N. (2017). Study of in vitro bioactivity and mechanical properties of diopside nano-bioceramic synthesized by a facile method using eggshell as raw material. Materials Science and Engineering: C, 71, 604-610.

-Najafinezhad, A., Abdellahi, M., Ghayour, H., Soheily, A., Chami, A., & Khandan, A. (2017). A comparative study on the synthesis mechanism, bioactivity and mechanical properties of three silicate bioceramics. Materials Science and Engineering: C, 72, 259-267.

Author Response

Please, see the attachment for the replies to the comments
